

# Cosmogenic nuclide weathering biases: Corrections and potential for denudation and weathering rate measurements

Richard F. Ott[1], Sean F. Gallen[2], Darryl E. Granger[3]

[1] GFZ German Centre for Geoscience Research, Potsdam, Germany
[2] Department of Geosciences, Colorado State University, Fort Collins, CO, U.S.A
[3] Department of Earth, Atmospheric, and Planetary Sciences, Purdue University, Purdue, IN, U.S.A.

*Correspondence to*: Richard F. Ott (richard.ott@gfz-potsdam.de)

**Abstract.** Cosmogenic radionuclides (CRNs) are the standard tool to derive centennial-to-millennial timescale denudation rates, however, it has been demonstrated that chemical weathering in some settings can bias CRNs as a proxy for landscape denudation. Currently, studies investigating CRN weathering biases have mostly focused on the largely insoluble target mineral quartz in felsic lithologies. Here, we examine the response of CRN build-up for both soluble and insoluble target minerals under different weathering scenarios. We assume a simple box model in which bedrock is converted to regolith at a

constant rate, and denudation occurs by regolith erosion and weathering either in the regolith or along the regolith-bedrock interface, as is common in carbonate bedrock. We show that weathering along the regolith-bedrock interface increases CRN concentrations compared to a no-weathering case, and how independently derived weathering rates or degrees can be used to correct for this bias. If weathering is concentrated within the regolith, insoluble target minerals will have a longer regolith residence time and higher nuclide concentration than soluble target minerals. This bias can be identified and corrected using

paired nuclide measurements coupled with knowledge of either the bedrock or regolith mineralogy to derive denudation and long-term weathering rates. Similarly, single nuclide denudation measurements can be corrected if a weathering rate and compositional data are available. Our model highlights that for soluble target minerals, the relationship between nuclide accumulation and denudation is not monotonic. We use this understanding to map the conditions of regolith mass, weathering, and denudation rates at which weathering corrections for cosmogenic nuclides become large and ambiguous as well as identify

environments in which the bias is mostly negligible, and CRN concentrations reliably reflect landscape denudation. We highlight how measurements of CRNs from soluble target minerals, coupled with bedrock and regolith mineralogy, can help to expand the range of landscapes for which centennial-to-millennial timescale denudation and weathering rates can be obtained.

## 1   Introduction

Denudation, the sum of physical erosion and chemical weathering, is a critical parameter to understand landscape evolution (Darwin, 1859; McLennan, 1993). Cosmogenic radionuclides (CRNs) have become the standard method to quantify denudation on centennial-to-millennial timescales and on spatial scales from outcrop to river basin (von Blanckenburg, 2005; Portenga and Bierman, 2011). The build-up of CRNs in minerals from eroding landscapes is sensitive to the rate of mass



removal from above (Lal, 1991). This makes the concentration of CRNs in the Earth's near-surface sensitive to physical erosion

as well as chemical weathering (Dixon et al., 2009; Riebe and Granger, 2013). However, weathering at depths below the attenuation length of CRNs, or measuring CRNs concentrations of minerals with a weatherability that differs from the bulk rock, can lead to biases in the measured CRN concentration (Dixon et al., 2009; Riebe et al., 2001; Riebe and Granger, 2013; Small et al., 1999).

Previous studies investigating the effects of chemical weathering on CRNs, hereafter just weathering, mostly focused on $^{10}$Be

measured on quartz in granitic landscapes (Dixon et al., 2009; Riebe et al., 2001, 2003; Riebe and Granger, 2013; Small et al., 1999). These studies show that quartz as a quasi-insoluble mineral is enriched in the saprolite and mobile regolith, whereas more weatherable minerals are depleted. This means that insoluble minerals reside longer in the mobile regolith compared to soluble minerals and the bulk rock. The longer residence time in the production zone leads to an increase in the quartz $^{10}$Be concentration that, without correction, results in an underestimation of the total denudation rate. However, the enrichment of

the insoluble mineral, commonly quantified by the ratio of zirconium in the regolith/saprolite compared to the bedrock, allows correction of the denudation rates for weathering (Dixon et al., 2009; Riebe et al., 2001; Riebe and Granger, 2013). This conceptualization of weathering biases has been developed and applied in granitic landscapes but has not been expanded to CRN measurements on the soluble target mineral, nor other weathering processes as they might occur in other regions subjected to weathering, e.g., limestone areas.

Carbonate regions are underrepresented in tectonic geomorphology studies despite representing ~ 18% of Earth's surface (Dürr et al., 2005), partially because measuring $^{10}$Be in quartz-rich lithologies has become the standard for catchment average denudation rate measurements (von Blanckenburg, 2005). Yet, carbonate regions are interesting because their topography is highly sensitive to the interplay between tectonics and climate, owing to their greater chemical reactivity relative to more silica-rich rocks, and they exert a significant influence on flora and fauna distributions (Ott, 2020). The number of studies

measuring $^{36}$Cl on calcite in carbonate catchments for denudation rates is increasing (Avni et al., 2018; Ott et al., 2019; Ryb et al., 2014a, 2014b; Thomas et al., 2017), and therefore the effect of weathering on soluble target mineral CRN concentrations needs to be assessed. This effort is worthwhile because it will allow extension of tectonic geomorphology and centennial-to-millennial timescale denudation rate studies in a significant portion of the globe that has been traditionally understudied. Moreover, the sensitivity of cosmogenic nuclide measurements to the weathering rate and depth will, in theory, allow for

measurements from minerals with different weatherability to constrain long-term weathering rates and depths in various geologic settings.

In this study, we investigate the theoretical effects of weathering on CRN concentrations in regions of substantial weathering, e.g., limestone areas, as well as how paired nuclide measurements can be exploited to constrain denudation and weathering rates on millennial timescales. We investigate the effects of weathering within the regolith, as well as preferential weathering

along the regolith-bedrock interface, on the CRN concentrations of soluble and insoluble target minerals. We derive the equations for the different weathering corrections and outline which practical measurements, for instance, water chemistry, can be used for the corrections. We highlight the non-monotonic relationship between nuclide concentration and denudation



rate for near-surface weathering of soluble target minerals. Carbonate landscapes can have a high ratio of weathering to erosion, which can lead to large correction factors. Therefore, we identify the conditions under which denudation and weathering rates

can be estimated with acceptable uncertainty and the environments where detailed data is needed to correct for the chemical weathering bias. This manuscript is accompanied by the code package "WeCode" (**We**athering **Co**rrections for **de**nudation rates) (Ott, 2022) integrated within the CRONUScalc v2.1 (Marrero et al., 2016, hereafter simply referred to as CRONUS) software to perform all weathering corrections and calculations, as well as offering pixel-by-pixel catchment production rate estimates for alluvial samples.

## 75    2    Summary of previous work

Small et al. (1999) noted the effect of near-surface weathering on the concentration of $^{10}$Be in quartz through the enrichment of weathering resistant quartz in the regolith compared to the bedrock. Riebe et al. (2001) showed that the concentration ratio of immobile elements such as [Zr] in the regolith and bedrock could be used to account for these weathering biases, assuming that the enrichment of [Zr] is the same as for quartz. Dixon et al. (2009) highlighted that weathering below the CRN attenuation

length would also underestimate the total denudational flux because the CRNs are only sensitive to near-surface mass removal. They show that total denudation rates from granitic areas in the Sierra Nevada underestimate the total denudational flux due to deep weathering by up to a factor of three. Riebe and Granger (2013) integrate the effects of weathering above and below the attenuation length with a box model of bedrock, saprolite, and soil and show that denudation rates in Puerto Rico were biased by up to 70% in comparison to a no-weathering assumption.

In this study, we follow Reibe and Granger (2013) and use a simple box model to quantify and account for the weathering bias on CRN concentrations. Because we focus on the application of carbonate terrains, we will neglect saprolite and instead apply a model of bedrock overlain by a layer of mobile regolith. Riebe and Granger (2013) show that in such cases, the target mineral, depending on its weatherability, gets either depleted or enriched in the regolith compared to the bulk rock; this will alter the regolith residence time and thereby the average nuclide concentration as follows:

$$< N_R > = N_0 + N_{R,nW} * \frac{X_R}{X_B} \ , \tag{1},$$

where $< N_R >$ is the average CRN concentration in a well-mixed regolith, $N_0$ is the nuclide concentration entering the regolith from the bedrock below, $N_{R,nW}$ is the nuclide concentration accumulated in the regolith in a no weathering scenario, and $X_R / X_B$ is the ratio of the target mineral in the regolith compared to the bedrock, which quantifies the change of mineral residence time compared to the bulk regolith. The aforementioned studies focused on the enrichment of quartz as an insoluble mineral. In this

study, we also explore the effects on soluble target minerals, such as calcite, for application to carbonate landscapes.



## 3    Theory and model

Some carbonate regions exhibit weathering mostly along the regolith-bedrock interface (Fig. 1A), whereas in other regions, weathering is focused within the regolith (Fig. 1B). Therefore, we investigate both weathering within the regolith and along the regolith-bedrock interface. We refer to regolith as the entire mobile layer above the coherent bedrock. We neglect
weathering below the attenuation length of CRNs because experiments have shown that majority of dissolution in limestone regions occurs close to the surface or at the top of bedrock (Gunn, 1981) and that deep mass removal is comparatively negligible in relation to near surface weathering (Worthington and Smart, 2004). First, we show the equations for the CRN concentration of a target mineral in the standard case of no weathering; second, we show the effect of regolith-bedrock interface weathering; finally, we highlight the impact of regolith weathering on CRN concentrations.

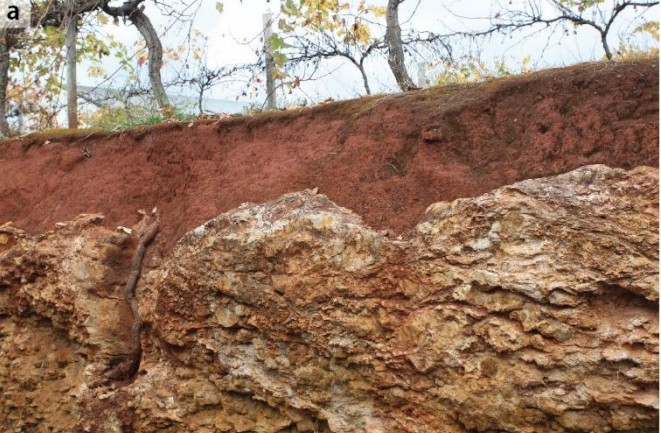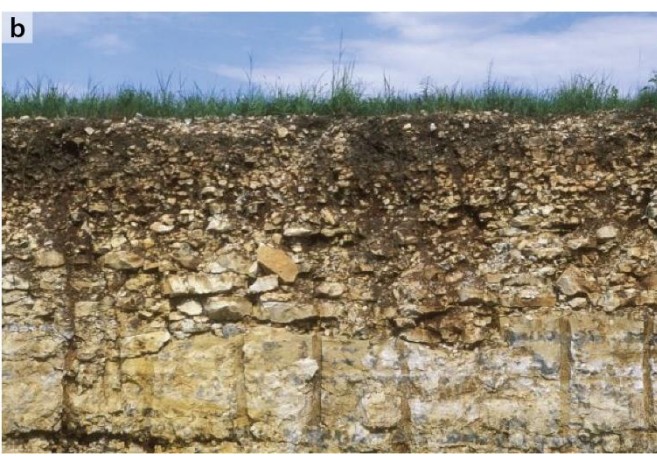

**Fig. 1: Weathering can be focused at different depths below surface. (a) Australia. Weathering focused mainly at the sharp regolith-bedrock interface. The soil consists mostly of residuum from the limestone bedrock together with aeolian dust. (b) Kansas, USA. Clasts released from the limestone bedrock at Flint Hills gradually decrease in size towards the surface. No clear regolith-bedrock interface is visible, indicating that weathering likely concentrates within the regolith.**



## 3.1    Model conceptualization and no weathering case

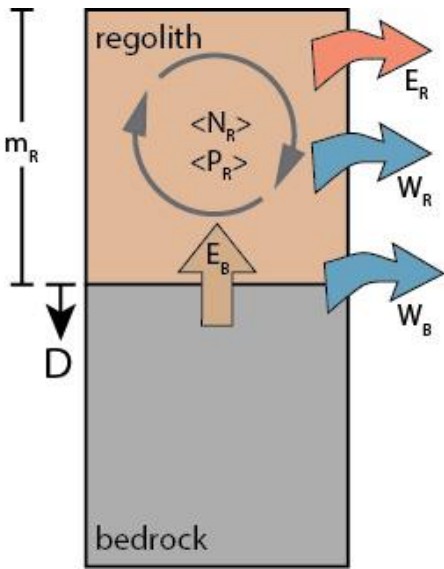

**Figure 2: Conceptualization of denudation as a simple box model.**

Analogous to previous studies, we conceptualize denudational processes with a box model comprising a layer of mobile regolith on top of unweathered bedrock (Fig.2). The bedrock weathers either at the regolith-bedrock interface with the bedrock weathering rate $W_B$, or gets converted to regolith at the bedrock erosion rate $E_B$. The regolith is assumed to be well-mixed and therefore has an average production rate $<P_R>$ and nuclide concentration $<N_R>$. Material is exported from the regolith either by regolith erosion ($E_R$) or weathering within the regolith ($W_R$). For simplicity, we assume a flux steady-state between denudation (D) and the outgoing fluxes ($W_R$, $E_R$, $E_S$), which results in a constant regolith mass ($m_R$) through time. Table 1 lists and defines all model variables.

The production rate of cosmogenic nuclides declines with depth, and the individual production profiles of different production pathways (spallation, fast muons, negative muons, etc.) are commonly approximated by exponentials (Braucher et al., 2011, 2013; Granger and Smith, 2000). Exponential production profiles for muons can result in significant biases for certain denudation rate calculations (Balco, 2017) but have the advantage of being simple in their mathematical formulation. The "WeCode" software package utilizes a full depth integration of the muon flux (Marrero et al., 2016); however, for demonstration purposes, we show equations using exponentials in section 3.1. and 3.2. For simplicity, we ignore radioactive decay. For $^{10}$Be, with a half-life of 1.36 Ma, this is acceptable for residence times shorter than a few hundred thousand years. For $^{36}$Cl, with a half-life of 301 ka, ignoring radioactive decay is somewhat more restrictive, requiring residence times shorter than ~100 ka.



**Table 1: Variable definition**

| Symbol | Name | Unit |
|:---:|:---|:---:|
| N | CRN concentration | at/g |
| $D$ | denudation rate (total mass loss) | g/cm²/ka |
| $E_B$ | bedrock erosion rate | g/cm²/ka |
| $E_R$ | regolith erosion rate | g/cm²/ka |
| $W_B$ | regolith-bedrock interface weathering rate | g/cm²/ka |
| $W_R$ | regolith weathering rate | g/cm²/ka |
| $\Lambda$ | attenuation length | g/cm² |
| $\rho$ | density | g/cm³ |
| $m_R$ | regolith mass | g/cm² |
| $P$ | production rate | at/g/ka |
| $i$ | subscript for production pathway | |
| $X_B, X_R$ | mineral fractions in bedrock and regolith | unitless |
| $Ca_B, Ca_R$ | fraction of calcite in bedrock and regolith | unitless |
| $X_B, X_R$ | fraction of non-quartz insolubles in bedrock and regolith (e.g. clay) | unitless |
| $Q_B, Q_R$ | fraction of quartz in bedrock and regolith | unitless |
| $t$ | time | ka |
| $\tau_R$ | average regolith residence time | ka |

### 3.1.1 No weathering case

In the absence of weathering, the weathering fluxes are zero such that $W_R$ and $W_B = 0$ and $D = E_B = E_R$. The average

nuclide concentration in the regolith is the sum of the nuclide concentration at the regolith-bedrock interface and the production within the regolith. This results in:

$$< N_R > = N_0 + < P_R > * \tau_R \tag{2}$$

with:

$$N_0 = \sum_i \left( \frac{P_i(0)\Lambda_i}{D} \right) e^{-m_R/\Lambda_i} \tag{3}$$

describing the nuclide concentration at the regolith-bedrock interface, and with:

$$< P_R > = \sum_i P_i(0) \left( \frac{\Lambda_i}{m_R} \right) (1 - e^{-m_R/\Lambda_i}) \tag{4}$$

describing the average production rate within the well-mixed regolith, and the regolith residence time:





$$\tau_R = \frac{m_R}{D}. \tag{5}$$

### 3.2 Weathering at regolith-bedrock interface

If weathering occurs exclusively at the regolith-bedrock interface, the denudation rate will be $D = E_B + W_B$. Sediment comes into the regolith at flux $E_B$ with the same concentration $N_0$ as in the no-weathering case. Therefore, the sediment flux into and out of the regolith goes down due to weathering at the regolith-bedrock interface. The modified regolith residence time is:

$$\tau_R = \frac{m_R}{E_B} = \frac{m_R}{D - W_B}. \tag{6}$$

This expression states that $\tau_R$ increases by a factor of $D_B/E_B$ compared to the no-weathering case. At steady-state, the regolith CRN concentration is:

$$<N_R> = N_0 + <P_R> * \frac{m_R}{D - W_B}. \tag{7}$$

Because $E_B < D$, the CRN concentration increases compared to the no-weathering scenario. Figure 3 shows this behavior for different regolith masses and fractions of regolith-bedrock weathering with respect to the total denudation rate, $D$. For

weathering fractions of 30%, the bias in CRN concentration compared to a no-weathering scenario is less than 25% for all regolith masses. However, for thick regolith, where most of the denudation is concentrated along the regolith-bedrock interface, the corrections become larger than 100% (Fig. 3). It is important to note that there is no difference between the denudation rate predicted by the soluble and insoluble target mineral in this simplified scenario. The insoluble mineral would be enriched within the regolith due to dissolution of the soluble target mineral at the interface, but without further regolith weathering, the

regolith residence time and resulting nuclide concentration would be the same for both mineral phases.

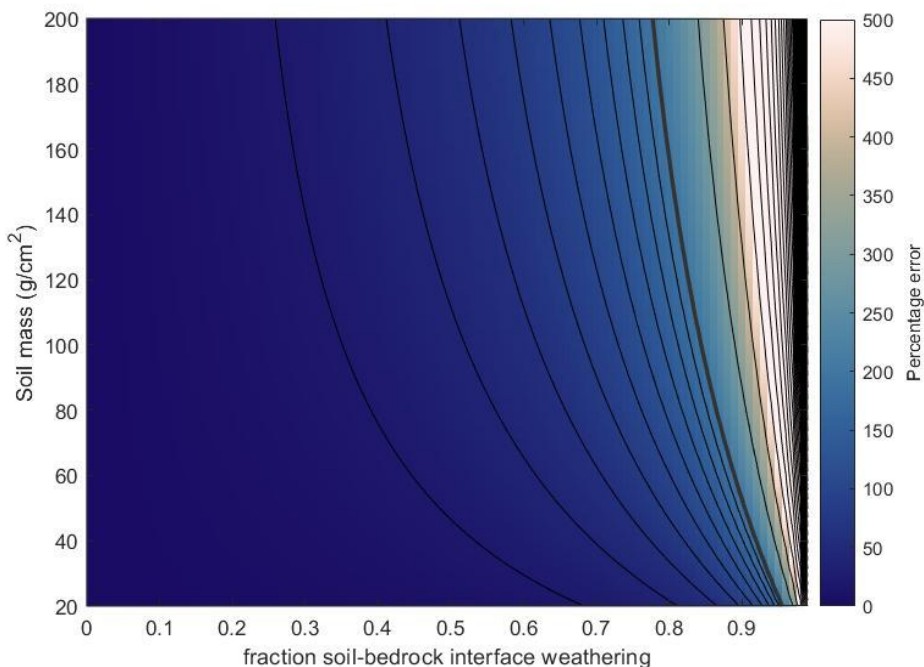

**Figure 3: The percentage of CRN concentration increase due to weathering at the regolith-bedrock interface compared to a no-weathering scenario. The percentage of CRN increase is plotted for different fractions of regolith-bedrock weathering in respect to denudation and different regolith masses. Contour lines every 20%; above 200% (thick line), the contours are every 100%.**

## 3.3    Regolith weathering

### 3.3.1    Homogeneous bedrock case – grain size effects

Consider a homogeneous soluble bedrock where regolith weathering and erosion occur such that the denudation rate $D = E_B = E_R + W_R$. The residence time of a parcel of rock within the regolith stays identical to the no-weathering scenario, $\tau_R = \frac{m_R}{D}$. The average regolith nuclide concentration will not be affected by regolith weathering because no parameter in Eq. 2 would change by partitioning the outgoing regolith fluxes differently. Despite the average nuclide concentration staying the same, regolith weathering would affect the size of grains in the regolith. Grains will lose mass with time in response to dissolution. A simple way to approximate weathering is as a mass loss proportional to grain mass such that:

$$m_g(t) = m_i * e^{-kt} , \tag{8}$$

where $m_g$ is the mass of a grain at time t, $m_i$ is the initial grain mass entering the regolith, and $k$ – a weathering constant such that $\frac{\partial m_g}{\partial t} = m_g * k$. While it may be preferable to use an expression that is proportional to surface area rather than mass, Eq. 8 captures the same general behaviour, but is more convenient because all variables are defined in terms of mass. If weathering is assumed to be proportional to grain mass, based on Eq. 8, grains never fully weather away. In nature, grains eventually disappear at some point, however, this common formulation (e.g., Gabet and Mudd, 2009) captures the general behaviour of



older grains contributing less mass and generally, the contribution of infinitesimally small grains to the total concentration is
negligible. A grain can only leave the regolith through regolith erosion $E_R$. Therefore, the average time a grain spends within the regolith becomes:

$$\tau_g = \frac{m_R}{E_R} .$$

(9)

It is important to note that this average grain residence time is always longer than the residence time of a rock parcel. This conceptualization of weathering predicts that with an increasing percentage of regolith weathering, the average regolith grain
size reduces and the grain residence time increases. The average CRN concentration of the regolith remains constant because older grains with higher concentrations contribute increasingly less mass to the total regolith and the average residence time of mass ($\tau_R$) stays constant. Moreover, weathering of grains will introduce a relation between grain size and the CRN concentration, where smaller grains will, on average, have higher concentrations (Fig. 4A), which does not exist for a no-weathering scenario. Following the assumption that erosion in a well-mixed regolith can be conceptualized as random plucking
of particles, the average regolith grain mass can be described as:

$$< m_g >= m_i * e^{-k*\frac{m_R}{E_R}} = m_i * e^{-k*\tau_g} .$$

(10)

This relationship predicts that the CRN concentration is grain size-dependent, with larger grains having a below-average and smaller grains an above-average CRN concentration, and it can be solved to determine the relative residence time of a given initial and final grain mass, i.e., $\tau_g = -k^{-1} * ln\left(\frac{m_g}{m_i}\right)$. In practice, this means that a CRN sample would need to be
representative of the grain size distribution in the regolith to accurately represent the denudation rate. Combining equations 2 and 10, one can define the nuclide concentration of a grain with respect to its mass:

$$N_g = N_0 + < P_R > * \frac{ln\left(\frac{m_g}{m_i}\right)}{-k} .$$

(11)

In practice, the collection of an alluvial sample for CRN measurement typically involves the selection of a certain grain size range. Depending on the relation of the sampled grain size to the total grain size distribution in the regolith, the sample will
either have a lower or higher CRN concentration compared to the average regolith (Fig 4C). For instance, sampling the coarser end of the regolith grain size distribution will result in a lower concentration compared to the average regolith and higher inferred denudation rates. For cases where grain size decreases systematically with time, only a sample integrating all grain sizes would result in a nuclide concentration reflecting the average regolith nuclide concentration.

A similar reduction in CRN concentration with increasing grain size has been linked to the supply of deep-seated material, e.g.
from landslides (e.g., Brown et al., 1995; Belmont et al., 2007; Aguilar et al., 2014; Puchol et al., 2014). Other studies have discussed a grain size reduction during sediment transport and weathering-facilitated sediment breakdown in the regolith (e.g., Carretier et al., 2009; Carretier and Regard, 2011; Lukens et al., 2016; Sklar et al., 2017; Lupker et al., 2017). The case presented here is more similar to the latter phenomenon, and in our model the magnitude of this grain size bias increases with



the weathering intensity. Thick soils and a high fraction of regolith weathering compared to the total denudation leads to a
large grain size bias. Areas with thin soils or low weathering intensities have negligible grain size bias.

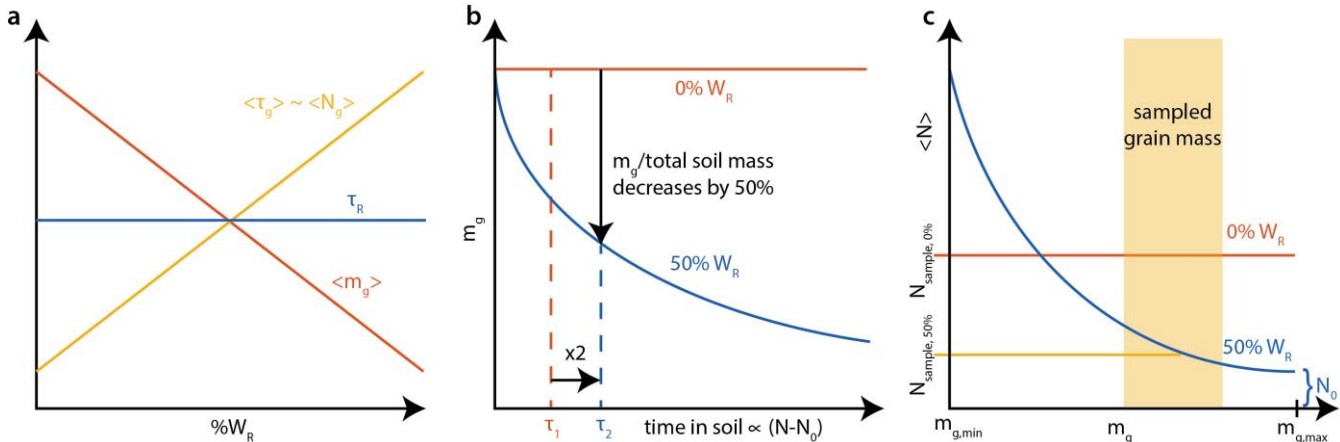

**Figure 4: Conceptual plots showing the effects of regolith weathering on a homogeneous soluble bedrock for weathering proportional to grain mass. (a) The average grain mass in the regolith <m_g> will decrease with increasing percentage of weathering $W_R$ if all other variables are fixed, while the average grain residence time $< \tau_g >$ and therefore grain CRN concentration would increase. However,**
**the average residence time of regolith mass $< \tau_R >$ will remain constant. (b) In the absence of weathering, no relationship is predicted between the grain mass and regolith residence time. Weathering will introduce such a relationship and lead to a decrease in grain size with time (blue line). If regolith weathering constitutes 50% of the denudation, the residence time of grain will double compared to the no-weathering case, however, the grain will also contribute 50% less to the total regolith mass. Therefore, no net difference in the average CRN concentration of the regolith occurs. (c) A CRN sample collected within a certain grain size range can be biased**
**due to weathering. In the case of regolith weathering, larger grain sizes will, on average, have a lower CRN concentration, that is, at minimum, the nuclide concentration entering the regolith $N_0$. Sampling the full regolith grain size range would result in the same CRN concentration compared to the no-weathering case. However, a sample from the coarser end of the grain size range (yellow box) would result in a too low CRN concentration.**

### 3.3.2 Regolith weathering in mixed bedrock mineralogy

Most carbonate-bearing rocks contain a mix of soluble and insoluble minerals. Let us consider the same regolith weathering scenario as in 3.3.1, but now with a bedrock containing minerals of heterogeneous weatherability. In this case, the soluble minerals would be depleted in the regolith, because they are removed by erosion and weathering, whereas the insoluble minerals, only affected by erosion, would be enriched (Riebe and Granger, 2013). The enrichment and depletion of minerals in the regolith involves a change of regolith residence time for the different minerals by a factor $X_R/X_B$ (concentration ratio of
mineral X in regolith, $X_R$, and bedrock, $X_B$). We can rewrite equation 1 for the average regolith CRN concentration measured in the target mineral X, using the more accurate depth integration formulation of the production rates instead of exponentials. In this case, the concentration of nuclides at the regolith-bedrock interface is the sum of nuclides produced through different production pathways $i$, integrated from the regolith residence time until infinity (past defined as positive values), such that:

$$N_{Bedrock\ Interface} = \sum_i \int_\tau^\infty P_i Dt\ dt .$$
(12)



The number of nuclides produced in the regolith is the integral of the average regolith production rate ($< P_{i,R} >$) from present to the average regolith residence time, modified by the enrichment/depletion factor ($X_R/X_B$), which accounts for a target mineral residence time that differs from the average. Hence, the average nuclide concentration of a target mineral $X$ in the regolith is:

$$< N_{R,X} > = \sum_i \frac{X_R}{X_B} \int_0^\tau < P_{i,R} > dt + N_{Bedrock\ Interface},$$ (13)

where the first term describes the nuclide accumulation within the regolith, and the second term is the CRN concentration from the advection of rock to the regolith-bedrock interface. This expression shows that measuring a nuclide concentration on a target mineral with a higher weatherability than the average bedrock will yield a CRN concentration lower than in a no-weathering scenario, and a less weatherable mineral will accumulate a greater nuclide concentration (Fig. 5). This occurs because the mean residence time for a soluble mineral decreases relative to the bulk regolith residence time, while the residence

time for an insoluble mineral increases relative to that of the bulk soil.

Following Riebe et al. (2003), we can write the mass balance for the mineral X as:

$$E_R * X_R + W_{R,X} = X_B * D_B \ ,$$ (14)

with $W_{R,X}$ as the regolith weathering rate of mineral X. Assuming the bedrock is a two-component mix of an insoluble mineral, I, and a soluble mineral, S. For the insoluble mineral, I, there will be no weathering ($W_{R,X} = 0$) and therefore enrichment factor

will be:

$$\frac{I_R}{I_B} = \frac{D}{E_R} = \frac{1}{1 - \frac{W_R}{D}} .$$ (15)

The insoluble mineral cannot be enriched to more than 100% of the regolith composition ($X_R = 1$). Therefore, we can define a minimum denudation rate $D_{min}$ that needs to be fulfilled to remain at a steady-state:

$$\frac{1}{I_B} = \frac{1}{1 - \frac{W_R}{D_{min}}} \quad \rightarrow \quad D_{min} = \frac{W_R}{1 - I_B} .$$ (16)

The depletion of the soluble mineral S is more complicated. The depletion factor $S_R/S_B$ depends on the concentration of the soluble mineral in either the regolith or bedrock. If we assume the bedrock concentration $S_B$ is known, we can rewrite Eq. 14 to:

$$\frac{S_R}{S_B} = \frac{S_B D - W_R}{S_B E_R} = \frac{S_B D - W_R}{S_B (D - W_R)} .$$ (17)

Equation 15 shows that the enrichment of the insoluble minerals is a function of the ratio of denudation to erosion, where

denudation needs to be operating at a rate greater than $D_{min}$. In contrast, Eq. 17 shows that the depletion of the soluble mineral is also a function of the bedrock composition. We will investigate the effects of this different behaviour in section 4.





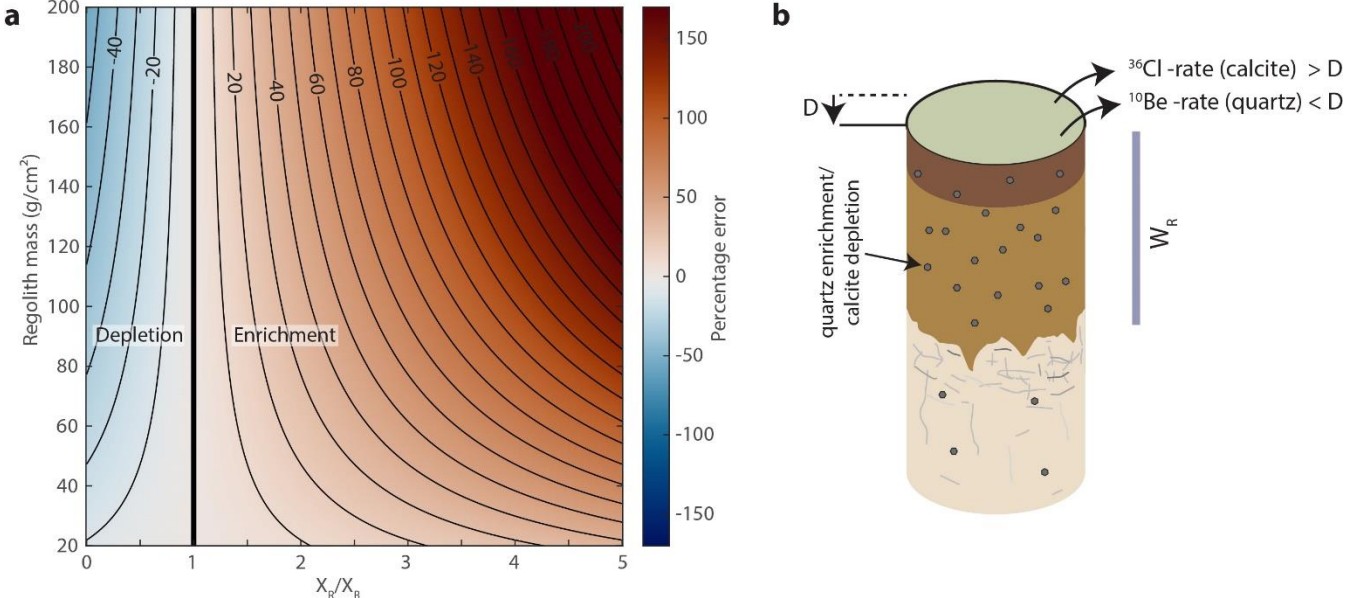

**Figure 5: (a) The percentage change in $^{10}$Be nuclide concentration compared to a no-weathering case for different regolith masses and enrichment/depletion factors at sea level high latitude (SLHL). (b) Conceptual representation of a rock with carbonate and**
**quartz minerals experiencing regolith weathering. The quartz grains would be enriched in the regolith compared to the calcite. Therefore, a $^{10}$Be denudation rate computed from quartz without correction would be lower than the average denudation rate, whereas a $^{36}$Cl denudation rate measured from calcite would be higher.**

## 4 Application of weathering corrections

In the previous section, we laid out the theoretical foundation for CRN weathering biases. Below we highlight how the above
equations can be used for practical weathering corrections in regions of non-negligible regolith-bedrock interface or regolith weathering. We outline how paired nuclide measurements from a soluble and insoluble target mineral can be used to calculate the landscape denudation rate as well as a long-term weathering rate. This method could provide a new tool to estimate long-term weathering. We also demonstrate how single nuclide measurements can be corrected for weathering. Previous studies performing regolith weathering corrections commonly employed the chemical depletion fraction (CDF) integrated over the
entire regolith (Dixon et al., 2009; Riebe et al., 2001; Riebe and Granger, 2013). Our study focuses on the application to limestone regions, where carbonate dissolution rates are commonly calculated from stream water chemistry. Therefore, we demonstrate how to use carbonate dissolution rates for the weathering corrections of CRN measurements.

The WeCode software performs all the calculations mentioned above, but also features the possibility to use the traditional CDF weathering correction. The production rate and scaling calculations are performed using CRONUS, and the input of
nuclide data follows the CRONUS input scheme. The current version of the code is written for $^{10}$Be in quartz and $^{36}$Cl in calcite since these are the most commonly measured insoluble and soluble target minerals, respectively. However, the code can easily be expanded to all other nuclides within CRONUS.





WeCode comes with a test data set to illustrate the application. The test data are supposed to reflect a typical "dirty" limestone composition of 70% calcite, 5% quartz, and 25% clay, where the clay component is regarded as insoluble on the timescale of
nuclide build-up. The denudation rate is 100 mm/ka, the calcite weathering rate 50 mm/ka, the regolith is relatively thick (200 g/cm²), and we use a mid-latitude low elevation location (48.585 °N, 9.250°E, 345 m). The scaling scheme for the test data is Stone (2000). For consistency and computational reasons, we use a linear error propagation scheme in line with CRONUS, where every parameter is varied by 10%, and the difference in the result is scaled by the uncertainty of the same parameter. For computational reasons, we only include uncertainty from CRN concentration, the weathering rate, and the spallation
production into the error estimation because we assume these to be the main sources of uncertainty. However, we acknowledge that the actual uncertainties might be greater and asymmetrically distributed and therefore not properly captured by our approach.

## 4.1 Regolith-bedrock interface weathering

Weathering at the regolith-bedrock interface affects the build-up of CRNs in the soluble and insoluble target mineral in the
same way. However, the mineral phases would be enriched or depleted directly at the interface, and hence the composition of the regolith would still follow Eqs. 15 and 17. If the weathering rate is known independently, e.g., from stream water chemistry or the weathering degree from a CDF, one can use equation 7 in section 2.2 to solve for the correct denudation rate. A CDF can be applied because it is related to the depletion of minerals in the regolith:

$$CDF = 1 - \frac{X_B}{X_R} \tag{18}$$

and, therefore, equation 7 can be adapted for the usage of a CDF to:

$$<N_R> = N_0 + <P_R> * \frac{m_R}{D} * \frac{1}{1-CDF}. \tag{19}$$

This correction requires that the majority of weathering is concentrated at the regolith-bedrock interface and the depth of this interface is known.

Regolith-bedrock interface weathering will lead to an increase in nuclide concentration and an underestimation of denudation
rate if not considered, independent of a measured target mineral. The CRN concentration of our test data predicts a conventional $^{10}$Be denudation rate of 61 mm/ka (assuming no weathering), but with 50 mm/ka weathering at the regolith-bedrock interface, the corrected denudation rate is 100 mm/ka (Fig. 6).

## 4.2 Regolith weathering

### 4.2.1 Paired nuclide measurements

The effects of regolith weathering can be corrected with paired nuclide measurements on a soluble and insoluble target mineral. Combining Eq. 13 of the nuclide build-up with the enrichment-depletion factors of Eqs. 15 and 17 demonstrates that the enrichment and depletion of the minerals depend on the total denudation, weathering rate, soil mass, and either the bedrock or





regolith composition. Knowledge of the soluble and insoluble mineral CRN concentrations, and either the regolith or bedrock composition, results in an equation system consisting of two CRN build-up equations and one equation relating the change in

composition to weathering (the exact equations would differ between providing either bedrock or regolith compositional data). This system of equations can be solved for the three unknowns of total denudation, weathering rate, and the mineralogy of the regolith or bedrock, respectively.

For accuracy, we use the standard CRONUS depth integration instead of the exponential production profiles introduced in section 3.1.1. Therefore, an analytical solution to the equation system is not possible, and we use an optimization algorithm to

solve for the correct combination of parameters. This approach predicts a weathering rate that integrates over the same time as the CRN measurements.

Carbonates typically contain a large amount of soluble calcite (or some aragonite or dolomite), a component of clay that can be regarded as insoluble on the timescale of calcite weathering and varying amounts of other minerals, such as, e.g., quartz. We can therefore expand the two-component system described in section 3.3.2 and add another insoluble component $X$,

representing, e.g., clays, such that:

$$Q_B + X_B = (Q_R + X_R) * \frac{E_R}{E_R + W_R} = (Q_R + X_R) * \frac{E_R}{D_R}, \tag{20}$$

$$\frac{Q_B}{X_B} = \frac{Q_R}{X_R}, \tag{21}$$

With Q representing the quartz fraction. Eq. 21 highlights that the ratio of the insoluble minerals stays constant between bedrock and regolith. Applying the above equations to our test data concentrations, we find that the uncorrected [10]Be

denudation rate would be 61 mm/ka, whereas the uncorrected [36]Cl denudation rate would be 132 mm/ka. Combining both CRN measurements with the bedrock composition yields a denudation rate of 100 mm/ka, with 50 mm/ka of weathering, where the mineralogy of the regolith is predicted from Eq. 20:

$$0.05 + 0.25 = (Q_R + X_R) * \frac{50}{50 + 50}, \tag{22}$$

as there is 10% quartz, 40% calcite, 50% clay in the regolith (Fig. 6). The erosion to denudation ratio of 1/2 predicts that quartz

and clay get enriched in the regolith by a factor of 2 compared to the bedrock, whereas calcite gets depleted by 57%.





**a - regolith-bedrock-weathering**

Measured:
$N_{10} = 64118 \pm 4300$ at/g
$W = 50 \pm 5$ mm/ka
Calculated:
$D = 100 \pm 7$ mm/ka

conventional: $D = 61$ mm/ka

$W_B$

**b - regolith weathering - single nuclide**

Measured:
$N_{36} = 216885 \pm 12100$ at/g
$W = 50 \pm 5$ mm/ka
Calculated:
$D = 100 \pm 19$ mm/ka
$X_{R,Q} = 0.10 \pm 0.02$
$X_{R,X} = 0.50 \pm 0.14$
$X_{R,Ca} = 0.40 \pm 0.12$

conventional: $D = 132$ mm/ka

Qz enrichment

$W_R$

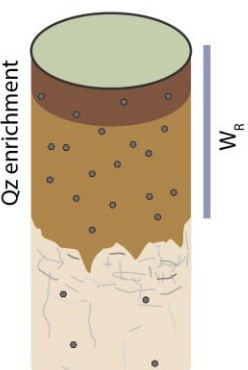

**c - regolith weathering - paired nuclide**

Measured:
$N_{10} = 64118 \pm 4300$ at/g
$N_{36} = 216885 \pm 12100$ at/g
Calculated:
$D = 100 \pm 6$ mm/ka
$W = 50 \pm 5$ mm/ka
$X_{R,Q} = 0.10 \pm 0.01$
$X_{R,X} = 0.50 \pm 0.04$
$X_{R,Ca} = 0.40 \pm 0.04$

conventional: $D_{10} = 61$ mm/ka
$\phantom{conventional: }D_{36} = 132$ mm/ka

Qz enrichment

$W_R$

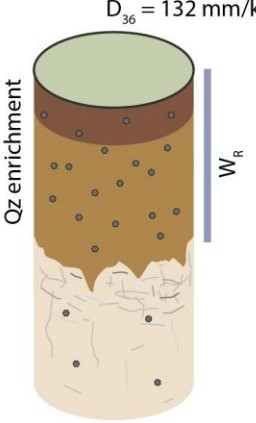

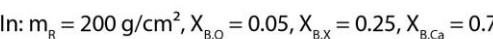

In: $m_R = 200$ g/cm$^2$, $X_{B,Q} = 0.05$, $X_{B,X} = 0.25$, $X_{B,Ca} = 0.7$

**Fig. 6: Calculation of "true" denudation rates for our test data. (a) Correction for weathering along the regolith-bedrock interface. The conventional denudation rate refers to a rate calculated without weathering correction. (b) Correction for a single nuclide measurement and regolith weathering. In this case the weathering rate, e.g. from stream chemistry needs to be known. (c) Calculation of denudation and weathering rate from a paired nuclide measurement, where one target mineral is insoluble and one is soluble. In this case we show the data for a combined measurement of $^{36}$Cl on calcite and $^{10}$Be on quartz.**

### 4.2.2 Single nuclide measurement

*Insoluble target mineral*

Previous studies focused on using an integrated CDF for weathering corrections on quartz (Dixon et al., 2009; Riebe et al., 2001; Riebe and Granger, 2013), which we also include in our code package. However, we highlight how the weathering correction can also be achieved with a weathering rate, e.g., independently derived from runoff and water chemistry measurements. This may be desirable in carbonate regions, where such weathering rate measurements are common. However, our corrections consider weathering within the cosmogenic nuclide production zone, and additional corrections are required for deep weathering, as observed, e.g., in tropical settings (Campbell et al., 2021; Dixon et al., 2009).





The enrichment of the insoluble target mineral only depends on the ratio of denudation to erosion, and therefore the weathering rate and the CRN concentration are sufficient for calculating a denudation rate. This becomes apparent if we combine the simplified cosmogenic nuclide concentration Eqs. 2 and 15:

$$< N_{R,X} >= N_0 + < P_R > * \frac{m_R}{D} * \frac{D}{E_R} = N_0 + < P_R > * \frac{m_R}{E_R} = N_0 + < P_R > * \frac{m_R}{(D - W_R)} \; . \tag{23}$$

This is the same as Eq. 7 in the regolith-bedrock interface weathering except for $W_R$ replacing $W_B$. In our synthetic data test

with 50 mm/ka weathering, the actual denudation rate would be 100 mm/ka in comparison to the conventional $^{10}$Be rate of 61 mm/ka. Similarly, to the paired-nuclide case, the mineralogy of the regolith can be predicted if the bedrock mineralogy is known and vice-versa. It is important to note that for the insoluble target mineral, an increase in denudation rate, while the weathering rate remains constant, will always lead to a decrease in CRN concentration. In equation 23, $N_0$ will decrease with increasing denudation, as well as the second term. The relationship between average nuclide concentration in the regolith and

denudation rate is monotonic.

*Soluble target mineral*

The depletion of the soluble target mineral depends on the mineral composition of the bedrock or regolith, additionally to the denudation and weathering rate. For the soluble target mineral, this leads to the counterintuitive behaviour where for a constant weathering rate, the CRN concentration does not necessarily decrease as denudation increases. If we combine the simplified

nuclide Eqs. 2 and 17 in the same manner as above, we get:

$$< N_{R,X} >= N_0 + < P_R > * \frac{m_R}{D} * \frac{S_B D - W_R}{S_B(D - W_R)} = N_0 + \frac{< P_R > m_R * \left(1 - \frac{W_R}{S_B D}\right)}{D - W_R} \; . \tag{23}$$

An increase in denudation will lower the concentration $N_0$, whereas depending on the composition, the regolith residence time may either become shorter due to an increase in the denudation flux, or may increase because the soluble target mineral gets less depleted. For denudation rates just above $D_{min}$, the depletion factor $X_R/X_B$ will increase rapidly (less depletion) and

therefore lead to an increase in CRN concentration until the maximum possible CRN concentration ($N_{max}$) is reached at denudation rate $D_{Nmax}$ (Fig. 7a). At denudation rates above $D_{Nmax}$ the increase in the regolith denudational flux outcompetes the lower depletion, and CRN concentration decreases (Fig. 7a). If we only consider CRN production in the regolith and neglect $N_0$, which applies to the case of thick regolith where production in the bedrock is negligible, we can use Eq. 23 to find the denudation rate with the maximum nuclide accumulation. Taking the derivative of Eq. 23 and setting it to zero returns the

maximum denudation rate by solving:

$$\frac{\partial < N_{R,X} >}{\partial D} = - \frac{< P_R > * m_R * \left(S_B D^2 - 2 W_R D + W_R^2\right)}{S_B D^2 * (D - W_R)^2} = 0 \; , \tag{24}$$

with the two solutions:

$$D_1 = \frac{W_R + W_R\sqrt{1 - S_B}}{S_B} \; and \; D_2 = \frac{W_R - W_R\sqrt{1 - S_B}}{S_B} \; , \tag{25}$$

where only $D_1$ is valid because $D_2 < D_{min}$.



It follows that for some nuclide concentrations, a soluble target will have two denudation rate solutions (see Fig. 7a). If the measured nuclide concentration is below the one expected for $D_{min}$, there will be a unique solution. We name the denudation rate above this threshold $D_{unique}$ and discuss the implications of this behaviour below.

The WeCode package checks the input data from soluble target minerals and notifies the user if there is more than one solution to a nuclide concentration. For non-unique solutions the code computes both denudation rates. We also provide functions to

calculate $D_{unique}$, $D_{Nmax}$, and $N_{max}$ for a given set of sample parameters. This can be used in advance to determine whether a sampling location is suitable to resolve the correct denudation rate from a soluble target mineral CRN measurement (e.g., [36]Cl).

## 5     Discussion

We investigated the effects of weathering on soluble and insoluble target mineral CRN concentrations for different weathering scenarios. We found that potential corrections for CRN denudation rate calculations exist, but additional data are required (i.e.,

a second isotopic system or independent weathering rate or CDF measurements) beyond a single CRN system measurement. In all cases, one needs an estimate of regolith thickness or mass, and the magnitude of the corrections increases with increasing regolith mass. We also show that paired nuclide measurements offer the possibility to constrain both denudation and long-term weathering rates.

Weathering along the regolith-bedrock interface is straightforward to correct for when an independent weathering rate is

known. The practical problem is to estimate where in the weathering zone the majority of weathering is taking place. Water chemistry measurements by Gunn (1981) show that most dissolution in carbonates takes place within the first meters below ground. A qualitative assessment of whether weathering occurs mostly in the regolith or along the regolith-bedrock interface could be based on regolith profiles in the field, analogous to figure 1, where a gradual grain size reduction in the profile indicates a dominance of regolith weathering.

The insoluble target mineral is blind to the difference between regolith-bedrock weathering and regolith weathering. Therefore, measuring the insoluble target mineral would be a simple way of avoiding ambiguity in regard to the weathering scenario. In contrast, the soluble target minerals will be affected differently depending on the weathering scenario; regolith-bedrock interface weathering would increase the nuclide concentration, and regolith weathering would decrease the nuclide concentration. Therefore, a paired nuclide measurement can, in theory, distinguish between the two scenarios. If the soluble

and insoluble target mineral CRN concentrations result in the same uncorrected denudation rate despite weathering, the result would suggest weathering occurred mainly along the regolith-bedrock interface.

### 5.1     Grain size bias

Continuous weathering within the regolith should reduce the grain size of soluble material released from the bedrock. In section 3.3.1 we highlight that the procedure of selecting a grain size window for alluvial cosmogenic nuclide measurements may

therefore be problematic in areas with significant weathering, as the grain size will be a function of regolith-residence time.



In practice, this grain size bias would be difficult to correct for because it would require knowledge of the grain size distribution entering or within the regolith. If, for instance, the grain size distribution entering the regolith is known and combined with a weathering law (e.g., weathering proportional to grain surface area), one could calculate the bias introduced by measuring a certain grain size. A paired nuclide measurement combined with either an independently derived weathering rate, or knowledge

of the regolith and bedrock composition, would allow one to evaluate if there is a grain size bias. A grain size bias would manifest itself through a predicted weathering rate that is higher than the independently derived one and a higher enrichment of insoluble minerals in the regolith than measured. Generally, areas with thin soils and/or low weathering intensity should not be affected strongly by grain-size biases.

## 5.2  Regolith Weathering

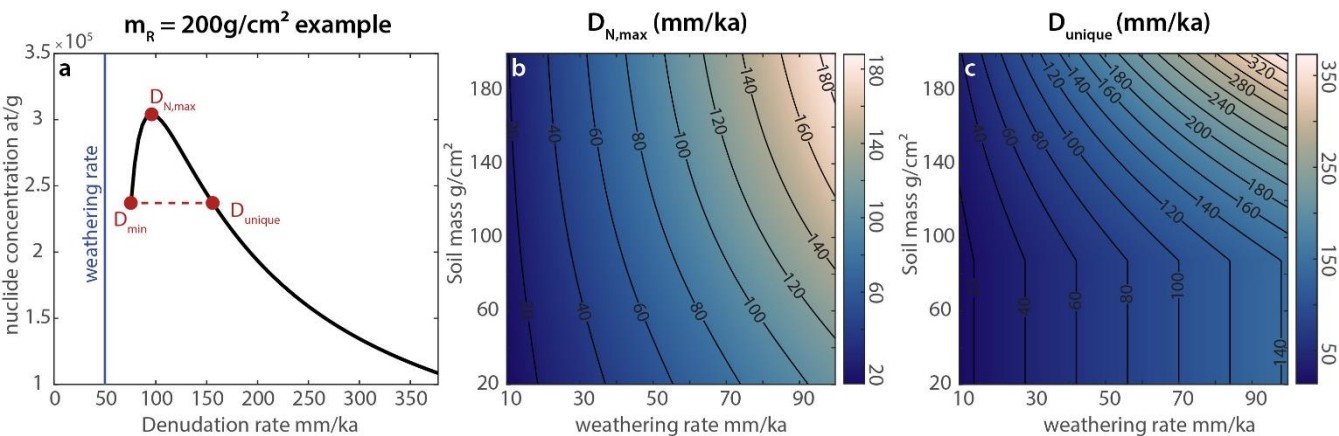


**Fig. 7. Weathering impact on the nuclide concentration of $^{36}$Cl measured in pure calcite at SLHL. $Ca_B$= 0.7, $Q_B$ = 0.3, $m_R$ = 200 g/cm². (a) $^{36}$Cl concentration versus denudation rate for calcite. The nuclide concentration increases above $D_{min}$ until $D_{Nmax}$. At denudation rates higher than $D_{unique}$, $^{36}$Cl measurements would be unambiguous. (b) $D_{Nmax}$ for the same sample, and a range of weathering rates and regolith masses. (c) same as (b) for $D_{unique}$.**

We have shown that paired nuclide measurements of soluble and insoluble target minerals have the potential to resolve a denudation rate, as well as a weathering rate. The weathering rate derived from such measurements is on the timescale of the nuclide build-up, and hence a long-term estimate of chemical weathering within the regolith. The grain size bias and the magnitude of weathering along the regolith-bedrock interface should be negligible for the determination of correct denudation and weathering rates. It is worth noting that the findings here can be applied to various regions, with mineral-nuclide

combinations beyond the $^{10}$Be-quartz, $^{36}$Cl-calcite pair highlighted in this study, e.g., quartz-magnetite, quartz-feldspar, quartz-pyroxene, or magnetite-olivine.

An important consideration before measuring CRNs for denudation rates in landscapes with non-negligible weathering is how large the expected corrections and potential errors would be. Large weathering corrections will come with considerable uncertainty due to the simplicity of the models used for correction and should therefore be avoided. But what magnitude of

correction should be considered too large?





We propose to use $D_{Nmax}$ and $D_{unique}$ as a quantitative measure of parameter combinations for the soluble target mineral, below which the recovery of a meaningful denudation rate becomes difficult. Ideally, the denudation rate within a region, for a given regolith mass and weathering rate should be higher than $D_{unique}$ that the nuclide concentration results in a unique solution for denudation rate. In our synthetic example with thick regolith, the depletion factor of calcite for $D_{unique}$ ($X_R/X_B$) is 0.78, and hence the weathering correction of the nuclide concentration is only ~ 22%. The weathering correction for $D_{Nmax}$ is ~ 55% of the nuclide concentration in our synthetic data. To avoid ambiguity in the interpretation of calculated denudation rates as well as large weathering corrections, we advise measuring soluble target minerals in regions with denudation rates above $D_{Nmax}$, and ideally above $D_{unique}$. The optimal regions for accurate denudation rate calculations are, therefore, areas with high denudation rates, thin regolith, and low to medium weathering rates. The settings that will generally meet these requirements for carbonate rocks are areas of moderate to high relief and temperate to arid climates.

Single nuclide measurements can be corrected for weathering using independent weathering rates derived from water chemistry. Chemical weathering rates from water chemistry integrate over a substantially shorter timescale (hours to a few years) compared to cosmogenic nuclides. Hence, caution needs to be observed when combining the two methods. Climate models can be used to check if climatic conditions, such as precipitation and temperature, have changed significantly throughout the cosmogenic nuclide averaging window, and can help to assess whether water chemistry-derived weathering rates could be biased (Ott et al., 2019).

A small caveat of WeCode is that it does not compute separate $^{36}$Cl production rates for regolith and bedrock. The change in chemical composition from bedrock to regolith due to weathering might affect the production rates for thermal and epithermal neutrons through the change in the fraction of absorbed neutrons. Most likely, the water content will also be higher in the regolith compared to the bedrock, which would increase the fraction of absorbed neutrons and lower production rates. We did not incorporate separate production rates for bedrock and regolith to avoid calculating a large set of production rates and slow down the computation significantly. However, for samples with low natural [Cl] or high water content, the production of thermal and epithermal neutrons is low, and the bias in the calculations is likely negligible. Except for cases of thin regolith, we recommend using the regolith values because that is where the majority of the $^{36}$Cl is produced.

## 5.3 Other considerations and future research needs

For single nuclide measurements, site-specific parameters should be evaluated before choosing which target mineral to sample. Measuring the insoluble target mineral quartz offers the advantage of minimizing the potential for a grain size bias. However, for lithologies with low quartz content, the enrichment factor and thereby the weathering correction of quartz will be substantially higher as the depletion factor for the soluble minerals. For a rock with 5% quartz, 95% calcite, and a denudation to erosion ratio of 2 $\left(\frac{D}{E_R} = 2\right)$, the enrichment factor of quartz in the regolith would also be 2, whereas the depletion factor of calcite would be 0.95. The enrichment of quartz would result in a larger weathering correction, and thus it is preferable to measure the CRN concentration in the soluble target mineral. This being said, the soluble target mineral may be experience an



undesired grain size bias, introducing a relationship between grain size and nuclide concentration. Hence, for single nuclide measurements, the choice of target minerals should therefore be made based on site-specific assessments of bedrock

composition, regolith thickness, and the expected range of weathering and denudation rates.

Another consideration should be whether the soluble and insoluble minerals are physically separated during the evolution of the regolith. If, for instance, insoluble minerals are part of larger rock fragments of soluble minerals while sitting in the regolith, both mineral phases should have the same residence time. If there is no physical mineral separation in the regolith, the denudation rates from soluble and insoluble minerals should be equal without weathering corrections.

For alluvial samples of soluble target minerals such as calcite, it is worth assessing whether secondary mineral precipitation occurs in the stream sediment. Perennial streams in limestone regions commonly form substantial amounts of secondary calcite within the sand fraction (Erlanger et al., 2021). The high solubility of [Cl] would result in low concentrations of [Cl] in the secondary minerals, however, experiments from cave waters suggest that the $^{36}$Cl signature of speleothems would be dominated by meteoric $^{36}$Cl (Johnston, 2010). Therefore, secondary target mineral precipitates would need to be identified

and, if relevant, removed from the sample before measurement.

Despite these caveats, weathering corrections of single nuclide measurements and paired nuclide measurements have the potential to expand the range of landscapes for which we can determine millennial timescale denudation rates. Several research needs are identified from our study that will ultimately assess the robustness of such weathering corrections. These research needs are listed as follows. (1) More measurements of cosmogenic nuclides with bedrock and regolith mineralogy are needed.

Such measurements will help to assess if the simple enrichment/depletion models for the evolution of the bedrock to regolith composition hold up in the field. (2) Paired nuclide measurements on different target minerals within the same sample are needed to test the divergence of nuclide concentrations. Especially, measurements from landscapes with an increasing degree of weathering to denudation ratio can be useful to test the hypothesis that the divergence in nuclide concentrations would increase, too. (3) Paired nuclide measurements combined with independent weathering rates can determine if the weathering

rates calculated from cosmogenic nuclides match other records. (4) More data on the distribution of weathering with depth, e.g., for carbonate regions, will be helpful to assess the relative importance of regolith and regolith-bedrock interface weathering in various landscapes. Cosmogenic nuclide measurements can contribute to this question because both weathering scenarios predict a different response of soluble target mineral CRN concentrations. (5) Potential grain size biases for soluble target minerals should be assessed. Paired-nuclide measurements can help to gauge grain size biases and estimate their

magnitude. Grain size measurements of the regolith and careful selection of sample grain sizes can be a way to quantify this potential bias.

The best conditions to test many of the hypotheses stated in this study will likely be met in Critical Zone Observatories. Unfortunately, few of them exist in limestone regions, highlighting the need for more data on regolith composition and evolution, especially in limestone regions of varying weathering rates. The same holds true for measurements of soluble target

minerals such as $^{36}$Cl in regolith and alluvial sediments. Most studies measure $^{36}$Cl on bedrock exposures (Avni et al., 2018; Godard et al., 2016; Matsushi et al., 2010; Stone et al., 1994; Thomas et al., 2018; Xu et al., 2013) and only a few in the regolith



or in alluvial sediments (Ott et al., 2019; Ryb et al., 2014b, 2014a; Thomas et al., 2017). Studying limestone regions and other areas with soluble minerals is of particular interest because the high weatherability makes them more susceptible to the interplay of tectonics and climate (Ott et al., 2019; Simms, 2004). More data on the weathering biases for cosmogenic nuclides
would allow expanding the calculation of denudation rates to new regions and improve our understanding of how the partitioning of denudation into erosion and weathering depends on tectonics and climate.

## 6    Summary and conclusions

We investigated the effects of chemical weathering on the nuclide concentration of soluble and insoluble target minerals in regolith-covered landscapes. Our main findings are:

(1)  In the case of regolith-bedrock interface weathering, independent knowledge of the weathering rate or degree from a CDF or water chemistry can be used to correct cosmogenic nuclide derived denudation rates independently from the target mineral weatherability.

(2)  In regions where weathering is concentrated within the regolith, paired nuclide measurements of a soluble and insoluble target mineral offer the potential to constrain the denudation rate as well as a long-term weathering
rate.

(3)  Previous studies have highlighted how single nuclide measurements on insoluble target minerals can be corrected using an integrated CDF value. We expand this approach and show that weathering rates from stream chemistry in combination with bedrock or regolith mineralogy can be used in the same way.

(4)  We derive equations that show the relationship between denudation and the nuclide concentration in soluble
target minerals is non-monotonous. We use this relationship to map the denudation rates $D_{Nmax}$ and $D_{unique}$, for various regolith masses and weathering rates; these can be used as guidelines for which areas to sample and which to avoid.

(5)  CRN measurements, especially from soluble target minerals, in combination with weathering rates, and compositional data from bedrock and regolith, can be used to assess the corrections proposed in this study.

## 7    Code availability

The        WeCode        software        package        is        available        at        https://dataservices.gfz-potsdam.de/panmetaworks/review/6070fc0104aed0a59bc47a2eda6260cf95ec1a09377fdf5bbc382a74cc52926f/ together with a user guide and the test data presented in this publication. The DOI provided in the references to Ott (2022) is reserved and will go online upon acception of this manuscript.





## 8    Author contribution

R.F.O., S.F.G., and D.E.G. designed the study. R.F.O. conducted the analysis, developed the WeCode software, and wrote the manuscript with input from all authors.

## 9    Competing interests

The authors declare no competing interests.

## 10    Acknowledgments

R.F.O. was supported by a contract from NAGRA (Swiss National Cooperative for the Disposal of Radioactive Waste) and by the by Swiss National Science Foundation fellowship, grant number P2EZP2_191866. We thank Jean Braun for discussions about this manuscript which aided our analysis.

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
