# Peer review of "Cosmogenic nuclide weathering biases: Corrections and potential for denudation and weathering rate measurements"

_Geochronology, 2022_

## Referee Comment (RC2)

[referee-annotated manuscript omitted]

---

## Author Response (AR1)

**Reviewer 1**

This is a fascinating and useful study of the effects of target solubility on CRN concentrations in a weathering substrate. The authors clearly present the theoretical underpinning and build up a model of target mineral enrichment/depletion in the weathered substrate. They additionally show that multiple weathering proxies can be used to correct for enrichment/depletion for basin-wide studies, which will be of broad interest. Using the WeCode package in advance of fieldwork will certainly be of benefit as well. Overall, this is a well-written manuscript and an excellent contribution to the field. I do have some concerns and points of clarification that I hope the authors will consider.

We thank the reviewer for the feedback and constructive comments that helped improve the manuscript and software package.

The basic conceptual model requires a mixed and eroding regolith. This is illustrated in figure 2, but it would be worth highlighting this and discussing the potential pitfalls that researchers might run into if this assumption is violated (i.e mixed but not eroding, or eroding but not well mixed).

This is a good point. Both scenarios (no denudation but mixing, no mixing but denudation) would not produce differences in nuclide accumulation between minerals of different solubility. In the case of "no denudation but mixing", the entire regolith would continue to accumulate nuclides at the rate of average production. If there is no mixing in the regolith, both minerals will accumulate nuclides, the same as in a no weathering scenario. There would be a density decrease due to the removal of a portion of the soluble target mineral. We have included a note in the discussion section 5.3 to address this comment:

"*If minerals are not physically separated in the regolith, the denudation rates from soluble and insoluble minerals should be equal without weathering corrections. The same should apply if no mixing or transport of the regolith occurs.*"

The variable k requires some extra discussion. It is defined on line 175 as a function of changing grain mass through time. I am guessing this is derived from the weathering rates? In any case, the manuscript could use a fuller discussion of how to derive k.

k is defined as the mass-loss rate of grains such that $\frac{\partial m_g}{\partial t} = m_g * k$; it is the rate constant. Hence, k, together with the grain size distribution in the regolith, determines the weathering rate. The weathering rate of the regolith could be obtained by integrating the differential equation above over all grain sizes, but requires assumptions about the grain size distribution. To illustrate this point in the manuscript, we added a sentence to section 3.3.1:

"*The regolith weathering rate $W_R$ can be related to k, by integrating the mass loss over all grain sizes, and hence, in practice, requires a known grain size distribution.*"

Line 124-126: why was radioactive decay ignored? As the authors point out, this is not likely a problem for $^{10}$Be, but $^{14}$C is becoming a common tool for quartz-based studies, especially in complex erosional settings where this technique will undoubtedly be used. Since the authors

must have used some form of the nuclide production equation, I would think it should not be too difficult to put the decay term back in.

We agree with the reviewer. In an effort to make the manuscript and the code more widely applicable, we have included decay in the revised version. The revised nuclide build-up equations, including radioactive decay, are harder to follow compared to the no-decay version. Since they produce the same general system behavior as the simpler no-decay equation, we decided to keep the no-decay version of the equations in the manuscript and add all equations with decay as a supplement. We have also included decay in the revised WeCode package.

We cite from the revised text:

*"We present the equations in the main text without decay because the equations with and without decay describe the same general behaviour, but the equations without decay are easier to follow. The equations with decay are presented in the supplement, and WeCode includes radioactive decay in all calculations."*

Table 1 is missing $m_g$, $m_i$, and k. (k is the most important)

We added the missing variables to Table 1.

Lines 183-185: I am struggling with the idea that the average grain residence time is always longer than the residence time of a parcel of rock. This is justified with equation 8, but this feels a bit like Zeno's paradox where you can never be shot with an arrow since it will always travel half the remaining distance. For soluble target minerals at low erosion rates, there must be a point where all that mineral has dissolved away before the original parent rock has moved through the regolith. It is possible that this is a function of the assumption of a mixed-eroding regolith. If so, that should be clearly stated.

The average residence time of a parcel of rock always has to be equal or longer than the average grain residence time of soluble minerals in a **homogeneous** bedrock. In case there is only erosion (i.e., no chemical weathering), both residence times will be equal. In a case where there is the same denudation rate, but it is partitioned into 50% erosion and 50% chemical weathering, mass is removed at the same rate as in the no weathering case, but the mechanical removal of grains by erosion is 50% slower. This behavior would also occur if the regolith were not mixed. The fact that in Eq.8, a grain will never fully weather away is a consequence of using a weathering law, where the mass loss is proportional to grain mass. In a low denudation, high weathering scenario, this would lead to some very small grains that have been sitting in the regolith for a long time. We agree that, in reality, many of these grains will fully dissolve instead of becoming infinitesimally small. However, for capturing the general behavior of grain size effects on the average regolith nuclide concentration, this does not matter because the small grains do not contribute a significant mass to the regolith. This was addressed in lines 176-180 of the original submission and figure 4.

It would be helpful to the reader to reiterate that $I_R$ and $I_B$ are mineral fractions and not concentrations.

We have added a note reiterating that it refers to the insoluble mineral fractions.

The unknowable errors that are mentioned on lines 290-292 are potentially very large. These could be partially addressed by expanding on the implications of the model assumptions as mentioned above.

We agree with the reviewer that this would be a valuable future research direction, and we highlight some of the ways of addressing this in lines 412-415, as well as the future research needs in lines 493-496 of the original submission. However, parameterizing these currently unconstrained uncertainties in, e.g. the grain size bias and including it in WeCode would be beyond the scope of this paper.

Line 307: 'corrected denudation rate' = input?

The weathering-corrected denudation rate would be the output. We have clarified this phrase.

Section 5.2 could be focused a bit more to remove repetition from earlier.

**Reviewer 2**

Ott et al. provide a solid theoretical framework for addressing weathering biases in denudation rate studies using soluble target minerals. This work expands on established literature that has been focused largely on insoluble minerals, and provides transparent, adaptable, and publically-available code that implements their approach. This work is timely, and includes a paired nuclide approach that should be useful for expanding into understudied landscapes. The manuscript is well-written, and the figures are (mostly - see below) clear and helpful.

We thank the reviewer for the positive and constructive feedback.

Re: Ignoring radioactive decay: How did you determine what was "acceptable" for the residence time thresholds given here? Is this the time over which ignoring decay would introduce a 5% error? More error than the analytical error? It feels a bit nitpicky, but being a bit more specific about what is deemed acceptable will help others implementing your code decide if ignoring radioactive decay is acceptable in their particular system. Especially for 36Cland 14C systems, decay could be worth considering in many systems. Adding it to the code seems like a relatively straightforward thing to implement, and would make the approach broadly applicable across a wider range of settings, including weathering studies outside of active tectonic settings. I certainly don't see adding decay to the code as a requirement for publication at this time, but it would be good to be clear about how big the impacts are for studies at the edges of the "acceptable" ranges given here.

We agree with the reviewer and this was also a point raised by reviewer #1. In an effort to make the manuscript and the code more widely applicable, we have included decay in the revision. The revised nuclide build-up equations, including radioactive decay, are harder to follow compared to the no-decay version and show the same behavior as the no-decay equations. We, therefore,

decided to keep the no-decay version of the equations in the manuscript and add all equations with decay as a supplement. We have also included decay in WeCode.

We cite from revised text:

"*We present the equations in the main text without decay because the equations with and without decay describe the same general behavior, but the equations without decay are easier to follow. The equations with decay are presented in the supplement, and WeCode includes radioactive decay in all calculations.*"

The discussion on possible grain size effects was very interesting. It makes sense to me that even without any grainsize-dependent sediment transport, size reduction of grains due to weathering and the associated range of particle residence times could introduce a relationship between particle size and CRN concentration. What's not clear from this conceptual framework is whether this effect will be large enough to worry about given all the other assumptions and sources of error. I'd love to see a full-blown model treatment to evaluate the possible magnitude of grain size effects in the context of other sources of error, but that's certainly beyond the scope here.

This is an interesting avenue for future research. The biggest unknown is the grain size distribution entering the regolith from the bedrock, and we simply don't have many (if any) measurements currently. If a distribution was known or assigned, one could use a weathering model, for instance, with a weathering rate proportional to grain mass as mentioned in section 3.3.1, to model the steady-state grain size distributions for different erosion-to-weathering ratios. This would allow one to estimate the nuclide concentrations of different grain sizes and hence the potential magnitude of the grain size bias, as described in section 5.1.

Minor comments:

In figure 5, it would be helpful to define $X_R/X_B$ in the caption, especially for readers who are skimming figures before they get into the meat of the text (or readers like me who get easily lost in variable alphabet soup when I'm tired)

We have adjusted the caption accordingly and added a definition of $X_R/X_B$.

Line 285: "regolith is relatively thick (200 g/cm$^2$)" – I assume from the units that this is an attenutation length, not a regolith thickness?

It is a soil mass per ground surface area. Assuming a density of 1.5 g/cm³ this would correspond to a 133 cm thick soil. We have adjusted this in the manuscript.

Figure 6: because the examples here use different nuclides, it would be easy to readers to be confused about the different between the scenarios. Just looking at the figure, it's easy to assume that scenario 1 is for 10Be, scenario 2 is for 36Cl, and scenario 3 is for both. Either 10Be or 36Cl would give the same result in (a), correct? You might consider just using 36Cl as the example in both (a) or (b), or hammering home that point in the caption.

We have adapted the caption to more specifically indicate that (a) could also be a measurement of a soluble target mineral.

Lines 374-375: "finding the denudation rate with the maximum nuclide concentration" and in the next sentence: solving for the maximum denudation rate – I'm confused, won't these be opposite (high CRN concentration = low denudation rate). Do you mean minimum D here? The notation around $D_{Nmax}$ is also a bit confusing, since it's a low denudation rate with "max" in the subscript. I think I understand why it was notated that way, but this section required super close reading to make sure I didn't get lost.

We have adjusted the second sentence to: "*Taking the derivative of Eq. 23 and setting it to zero returns the denudation rate with maximum nuclide concentration by solving:*"

We also added a sentence to help the reader follow the counterintuitive behavior of soluble target minerals. We cite from the revised manuscript:

"*In other words, at $D_{min}$, the soluble target mineral weathers away instantaneously upon entering the regolith and an increase of denudation will lead to a fraction of grains surviving thereby increasing the residence time and concentration; a further increase in denudation however, will lower the residence time and nuclide concentration again.*"

We understand that the variable name $D_{Nmax}$ is a bit counterintuitive in the sense that it is on the low end of permitted denudation rates while carrying the subscript max. However, it is the denudation rate D at which the maximum nuclide concentration N occurs. We, therefore, decided to stick with this naming convention because we cannot think of a better name.

There are a few minor typos and formatting things annotated in the attached PDF.

We thank the reviewer for pointing out these typos and have corrected them in the revised manuscript.